# Facial Emotion Recognition in Patients with Post-Paralytic Facial Synkinesis—A Present Competence

**DOI:** 10.3390/diagnostics12051138

**Published:** 2022-05-04

**Authors:** Anna-Maria Kuttenreich, Gerd Fabian Volk, Orlando Guntinas-Lichius, Harry von Piekartz, Stefan Heim

**Affiliations:** 1Department of Otorhinolaryngology, Jena University Hospital, Am Klinikum 1, 07747 Jena, Germany; fabian.volk@med.uni-jena.de (G.F.V.); orlando.guntinas@med.uni-jena.de (O.G.-L.); 2Facial-Nerve-Center Jena, Jena University Hospital, Am Klinikum 1, 07747 Jena, Germany; 3Center of Rare Diseases Jena, Jena University Hospital, Am Klinikum 1, 07747 Jena, Germany; 4Department of Psychiatry, Psychotherapy and Psychosomatics, Medical Faculty, RWTH Aachen University, Pauwelsstr. 30, 52074 Aachen, Germany; sheim@ukaachen.de; 5Department of Neurology, Medical Faculty, RWTH Aachen University, Pauwelsstr. 30, 52074 Aachen, Germany; 6Department of Physical Therapy and Rehabilitation Science, Osnabrück University of Applied Sciences, Albrechtstr. 30, 49076 Osnabrück, Germany; h.von-piekartz@hs-osnabrueck.de; 7Institute of Neuroscience and Medicine (INM-1), Forschungszentrum Jülich, Leo-Brand-Strasse 5, 52428 Jülich, Germany

**Keywords:** facial palsy, post-paralytic facial synkinesis, emotion recognition, facial feedback

## Abstract

Facial palsy is a movement disorder with impacts on verbal and nonverbal communication. The aim of this study is to investigate the effects of post-paralytic facial synkinesis on facial emotion recognition. In a prospective cross-sectional study, we compared facial emotion recognition between *n* = 30 patients with post-paralytic facial synkinesis (mean disease time: 1581 ± 1237 days) and *n* = 30 healthy controls matched in sex, age, and education level. Facial emotion recognition was measured by the *Myfacetraining* Program. As an intra-individual control condition, auditory emotion recognition was assessed via *Montreal Affective Voices.* Moreover, self-assessed emotion recognition was studied with questionnaires. In facial as well as auditory emotion recognition, on average, there was no significant difference between patients and healthy controls. The outcomes of the measurements as well as the self-reports were comparable between patients and healthy controls. In contrast to previous studies in patients with peripheral and central facial palsy, these results indicate unimpaired ability for facial emotion recognition. Only in single patients with pronounced facial asymmetry and severe facial synkinesis was an impaired facial and auditory emotion recognition detected. Further studies should compare emotion recognition in patients with pronounced facial asymmetry in acute and chronic peripheral paralysis and central and peripheral facial palsy.

## 1. Introduction

Facial palsy can affect the face in function and appearance [1] with various consequences [2]. Due to the motor impairment of the facial muscles and its effect on verbal and nonverbal communication, it can also be considered as a communication disorder [3]. Accordingly, there may be emerging constraints on verbal communication such as articulation and intelligibility [4], as well as in nonverbal communication [3]. In nonverbal communication, it can be difficult for patients with facial palsy to express facial emotions [5,6,7,8] and their conversation partners experience this impaired mimic communication [6]. The interlocutors perceive the patients’ deformed appearance negatively [9], and attribute negative emotion expressions even while smiling [10]. Further, the patients are judged significantly less likeable, less trustworthy [11], and less attractive [9,11]. This external perception may lead to social exclusion [11] and stigmatisation [3,12]. About 20–30% of the patients do not recover from facial palsy, thus continuously suffering from post-paralytic facial synkinesis [13,14,15,16,17,18,19], i.e., involuntary muscle movement while executing a different, intentional muscle movement [19].

Despite the fact that different negative consequences of facial palsy are already known, the communicative effects of facial palsy and post-paralytic facial synkinesis are not conclusively elucidated yet. For example, the performance of facial emotion recognition has received limited scientific attention so far, although the first indications of deficits are available in recent studies (details below) [20]. Up until today, as a review on emotion processing in patients with facial palsy summarised in the available studies, it is not finally clarified whether patients with facial palsy and especially those who suffer from post-paralytic facial synkinesis are severely and systematically affected by impaired facial emotion recognition [20]. 

Depending on the location of the lesion, peripheral and central facial palsy are distinguished [16]. For both types, the first evidence on facial emotion recognition is available: In a recent study, we examined the facial emotion recognition of patients with central facial paresis after a stroke. The patients demonstrated significant deficits in accuracy of facial emotion recognition when compared to patients after a stroke without facial paresis and healthy controls [21,22,23]. Although the results demonstrate a specific deficit in facial emotion recognition compared to auditory emotion recognition of patients with facial paresis, the cause for these limitations is not completely elucidated. Both stroke [24] and altered facial feedback [25] could influence facial emotion recognition. Consequently, facial emotion recognition should be tested in patients with altered facial feedback, e.g., with post-paralytic facial synkinesis, but without any neurological precondition [21,22,23]. 

Recent studies have also documented impaired facial emotion recognition in patients with peripheral facial palsy. For example, Storbeck et al. reported that 31 patients with acute peripheral facial palsy were significantly slower in facial emotion recognition in comparison to their healthy controls [26]. Moreover, Konnerth et al. presented similar results (significantly slower), when comparing 13 patients with chronic (>four weeks post onset) peripheral facial palsy and a healthy control group [27]. Korb et al. even identified impairments especially for patients with left-sided facial palsy [28]. However, overall, there is only a small number of studies that examine facial emotion recognition in patients with peripheral facial palsy [20].

Our present study addresses this open issue. To uncover existing deficits or competences, the aim of this study is to observe the effects of post-paralytic facial synkinesis on facial emotion recognition.

## 2. Materials and Methods

### 2.1. Study Design

In a prospective cross-sectional study, we examined patients with post-paralytic facial synkinesis and compared them to healthy controls in facial emotion recognition in accuracy and time. We also tested auditory emotion recognition to distinguish intra-individually general deficits in emotion recognition from modality-specific deficits in facial emotion recognition. Moreover, we assessed the participants’ subjective judgements of their own facial emotion recognition abilities. 

This selection and combination of measurements and assessments is already established and proven to be useful. It was developed in a recent study on facial emotion recognition focusing patients with central facial paresis. There, this set of measurements and assessments demonstrated reliable discrimination between patients with and without central facial paresis as well as healthy controls and uncovered deficits [21,22,23]. 

The study was conducted according to ethical standards and was approved by the ethical committee (2020-1787-BO) of the Jena University Hospital Jena, Germany. All participants signed a consent form voluntarily after they had been informed in detail about the study.

### 2.2. Participants

Two target groups of participants were recruited: (1) patients (adults ≥18 years) with unilateral, chronic (≥1 year post onset) peripheral facial palsy with related facial synkinesis, diagnosed by an expert physician, and (2) healthy controls (adults ≥18 years) with no history of facial palsy, and no acute or chronic facial palsy with or without facial synkinesis. All patients and healthy controls had normal or corrected vision and hearing ability assessed by the participant. None of the participants had a diagnosis of neurological or mental disorders and/or taking antidepressants.

Recruitment and data collection were conducted from 4 August 2020 to 20 August 2021. The patients were recruited at the Facial-Nerve-Center Jena, Jena University Hospital, Jena, Germany. The examinations took place either at the Jena University Hospital or during home visits if requested from the participants. None of the participants had ever received prior facial emotion recognition diagnostic or specific emotion recognition therapy.

A total of 30 patients with post-paralytic facial synkinesis and 30 healthy controls without facial palsy were included. The patients and the healthy controls were matched in pairs based on their sex, age, and education level. These three factors, sex, age, and education were selected because they may have an impact on physiological emotion recognition (sex: [29,30]; age: [31]; education level: [32]). Characteristics of the patients and the healthy controls are presented in Appendix A, Table A1. Most patients and healthy controls were female, middle aged, and with an education of medium maturity. There were no significant group differences between patients and healthy controls in sex, age and education level. Mean duration of facial palsy was 1581 ± 1237 days. Detailed information of facial palsy is provided in Appendix A, Table A2.

#### Diagnosis and Grading of Facial Palsy and Facial Synkinesis

In order to study the presence and grading of a possible facial palsy and facial synkinesis, all participants went through examination (conducted by a speech and language therapist) regarding facial palsy in addition to the diagnosis by a physician.

For this purpose, all participants were instructed in a standardised manner to show their face at rest and then to perform voluntary movements with their face. The examination was recorded with a video camera (CANON, HF100, Tokyo, Japan; camera at a right angle, positioned 150 cm away from the participant’s chin, camera lens at the individual level of the participant’s chewing plane) and graded according to the *Sunnybrook Facial Grading System* [33,34] afterwards. This tool is used to determine a composite total score (Composite Score; 0: total facial paralysis to 100: normal face) from sub scores for symmetry at rest (Resting Symmetry Score), voluntary movements (Voluntary Movement Score), and synkinesis (Synkinesis Score) [33,34]. In addition to this rating, to distinguish between faces with and without facial palsy and to ascertain the degree of facial palsy, we classified the composite total score of *Sunnybrook Facial Grading System* according the House & Brackmann *Facial Nerve Grading System* [35] into six degrees of severity: 100–84 points: normal function, no facial paresis/paralysis; 83–67: light facial paresis; 66–50 moderate facial paresis; 49–33 medium facial paresis; 32–16 severe facial paresis; 15–0 complete facial paralysis. Thus, a certain degree (16 points out of 100 points) of asymmetry was accepted as physiological.

Facial palsy was excluded in all healthy controls (Composite Score: Mean = 91.3 ± 4.5; Grade Median = 1). In all patients, facial synkinesis was confirmed (Composite Score: Mean = 39.4 ± 15.8). About half of the patients had medium facial paresis (Grade Median = 4). Abnormalities were observed in all patients through resting symmetry (Resting Symmetry Score: Median = 15), voluntary movement (Voluntary Movement Score: Median = 58), and synkinesis (Synkinesis Score: Median = 6). Thus, all patients had unilateral facial synkinesis [15,16,17,19]. About half of the patients were affected on the left side, while the other half of the patients were affected on the right side of their face. Detailed information on the diagnosis, affected side, grading, aetiology, and the time post onset of the facial palsy (patients), or facial integrity (healthy controls) can be found in Appendix A, Table A2.

### 2.3. Materials for Measuring Facial and Auditory Emotion Recognition and Self-Assessment of Facial Emotion Recognition

In order to test emotion recognition, the so-called basic emotions of anger, disgust, fear, joy, sadness, and surprise [36,37] were used. These emotions are considered as unambiguous and culture-independent [38] and are typically used for this purpose [39].

We examined all participants once. For measuring emotion recognition, the accuracy (percentage of correctly recognised items) and time (average speed in seconds) were recorded in facial and auditory modality. To ensure that the tasks were understood by the participants, a pre-test with ten items was presented beforehand. These procedures are explained in the following sections and has already been established and described in more detail before [21,22,23].

#### 2.3.1. Measuring Facial Emotion Recognition

Data of facial emotion recognition was collected using the *Myfacetraining* (MFT) Program (CRAFTA Cranio Facial Therapy Academy, Hamburg, Germany; [40,41]). Each participant rated 42 photographs of people showing one of the six basic emotions on their face. The participants were asked to choose the presented emotion from different options within ten seconds as correctly and quickly as possible (presented from a laptop Lenovo yoga 500 (Lenovo, Hongkong, China), screen size 14 inches, touch screen input mode) see also [21,22,23].

#### 2.3.2. Measuring Auditory Emotion Recognition

To test auditory emotion recognition, a selection (basic emotions) of the *Montreal Affective Voices* (MAV) [42] was presented. The participants were asked to rate a total number of 60 emotional, non-linguistic, vocal expressions (on/a:/) of the basic emotions by selecting the presented emotion from different options as correctly and quickly as possible. Each item was presented once and ten seconds responding time was given (input mode: point on symbols on a DIN-A4 paper sheet) through a specially programmed experiment in PsychoPy (version 3.0.0b9; [43]; laptop Lenovo yoga 500 (Lenovo, Hongkong, China), screen size 14 inches, using commercially available wired headband headphones with an individual volume at the participant’s discretion), see also [21,22,23].

#### 2.3.3. Self-Assessment of Facial Emotion Recognition

To self-assess facial emotion recognition, all participants filled out four questionnaires for two categories: (1)*Overall competence in accuracy and time:* Two questionnaires were developed to assess overall competence in facial emotion recognition on a visual analogue scale (10 cm). All participants were asked how accurate (questionnaire for accuracy) and how fast (questionnaire for time) they would rate themselves in recognising the six basic emotions in other people’s faces.(2)*Changes in accuracy and time since facial palsy:* Further, we adapted and used two questionnaires (one for accuracy, one for time) [21,22,23] to assess possible changes in facial emotion recognition since the onset of their facial palsy. The patients were asked whether they noticed changes in accuracy and time, when recognising each of the six basic emotions in other faces (−1 point: less accurate/slower; 0 points: no change; +1 point: more accurate/faster). Moreover, the healthy controls were requested if they noticed any changes over a comparable period of time (average time of facial palsy duration in patients: mean = 1581 days ≙ 4 years; see also Appendix A, Table A2).

### 2.4. Statistical Analysis

All data were analysed using Microsoft Excel 2019 (Redmond, WA, USA; [44] and IBM SPSS Statistics 28.0 (Armonk, NY, USA; [45]. Results at an alpha level of *p* < 0.05 were considered significant.

To compare the results of facial and auditory emotion recognition (accuracy and time) obtained from the patients and the healthy controls, 2 × 2 ANOVAs with *group* (patients vs. healthy controls) as between-subject factor and *modality* (facial vs. auditory) as within-subject repeated-measures factor were conducted. Significant interactions were resolved by post hoc *t*-tests for dependent samples to test facial vs. auditory emotion recognition (accuracy and time) separately within each group of participants.

In addition to this analysis of emotion recognition, factors were explored, which are systematically related to the performance of emotion recognition. If such factors could be identified, they may provide further insight on emotion recognition and could be considered in the evidence-based care of patients with facial palsy. Therefore, correlations for facial and auditory emotion recognition in accuracy and time (Pearson) as well as accuracy/time and sex (Pearson), age (Pearson), education (Spearman), overall severity of facial palsy (Pearson), and separately sub scores resting symmetry, voluntary movements, and facial synkinesis (Spearman) were calculated. Moreover, *t*-tests for independent samples to test the facial emotion recognition of patients with left- and right-sided facial synkinesis were run.

In a last step, the self-assessment of the patients and the healthy controls in emotion recognition (accuracy and time) were compared through *t*-tests for independent samples, and within each group of participants (patients or healthy controls) through *t*-tests for dependent samples. Moreover, correlations for self-assessed facial emotion recognition in accuracy and time (Pearson) as well as self-assessed and measured facial emotion recognition (Pearson) were conducted.

## 3. Results

The results of measured facial and auditory emotion recognition as well as computed ANOVAs and *t*-tests are presented separately by accuracy and time. Significant results are reported in text and all results (significant and not significant) are summarised in Table 1. In further analysis, announced correlations are reported. Finally, the results for self-assessment and its correlations are described.

### 3.1. Accuracy

The 2 × 2 ANOVA yielded a significant main effect of *modality*, with higher accuracy in facial emotion recognition than in auditory emotion recognition. 

No significant main effect of *group*, and no significant interaction effect of *group × modality* was identified.

Table 1 shows the results of measured emotion recognition and the statistical analysis. Figure 1 shows facial and auditory emotion recognition in accuracy as a function of time post onset.

### 3.2. Time

The 2 × 2 ANOVA yielded a significant main effect of *modality*, with longer response times in facial emotion recognition than in auditory emotion recognition.

No significant main effect of *group*, and no significant interaction effect of *group × modality* was identified.

Table 1 shows the results of measured emotion recognition and the statistical analysis. Figure 2 shows the facial and auditory emotion recognition in time as a function of time post onset.

Table 1 shows an overview of all results of the measured emotion recognition. No significant differences were found between patients and healthy controls, but within each participant group, depending on the tested modality.

### 3.3. Further Analysis of Measured Facial Emotion Recognition

For further analysis, we studied different possible correlations as described in the section of *Statistical Analysis.* All results in correlations (significant and not significant) are summarised in Table 2, Table 3 and Table 4. Significant correlations are also presented in the text.

#### 3.3.1. Correlation between Accuracy and Time in Measured Facial and Auditory Emotion Recognition

There was a significant positive correlation between accuracy for facial emotion recognition and accuracy for auditory emotion recognition for all participants as well as in the patients, and in the healthy controls.

There was a significant negative correlation between accuracy for facial emotion recognition and time for auditory emotion recognition for all participants, and in the patients.

There was a significant negative correlation between time for facial emotion recognition and accuracy for auditory emotion recognition in the healthy controls.

There was a significant positive correlation between time for facial emotion recognition and time for auditory emotion recognition for all participants as well as in the patients, and in the healthy controls.

#### 3.3.2. Correlation between Accuracy/Time of Measured Facial Emotion Recognition and Sex

There was a significant negative correlation between accuracy and sex across all participants, with higher accuracy for females.

#### 3.3.3. Correlation between Accuracy/Time of Measured Facial Emotion Recognition and Age

There was a significant negative correlation between accuracy and age across all participants (Figure 3).

There was a significant positive correlation between time and age across all participants (Figure 4).

Further, when examining the patients, there was a significant negative correlation between accuracy and age as well as a significant positive correlation between time and age. Similarly, the measures of the healthy controls correlated significantly negative between accuracy and age as well as significantly positive between time and age.

#### 3.3.4. Correlation between Accuracy/Time of Measured Facial Emotion Recognition and Education

There was a significant positive correlation between accuracy and education across all participants. The measures of the healthy controls demonstrated a significant positive correlation between accuracy and education.

#### 3.3.5. Correlation between Accuracy/Time of Measured Facial Emotion Recognition and Overall Grading of Facial Palsy

Figure 5 and Figure 6 show correlations between accuracy/time of measured facial emotion recognition and overall grading of facial palsy *(Sunnybrook Facial Grading System: Composite Score).*

#### 3.3.6. Correlation between Accuracy/Time of Measured Facial and Auditory Emotion Recognition and Facial Resting Symmetry

In the patients, there was a significant negative correlation between accuracy for facial emotion recognition and facial resting symmetry *(Sunnybrook Facial Grading System: Resting Symmetry Score).* Moreover, in patients, there was a significant negative correlation between accuracy for facial emotion recognition and resting symmetry mouth.

#### 3.3.7. Correlation between Accuracy/Time of Measured Facial and Auditory Emotion Recognition and Facial Synkinesis

In the patients, there were significant negative correlations between accuracy for facial emotion recognition and facial synkinesis *(Sunnybrook Facial Grading System: Synkinesis Score)* and synkinesis in snarl. Moreover, in patients, there was a significant positive correlation between time for facial emotion recognition and synkinesis in snarl.

In the patients, there was a significant negative correlation between accuracy for auditory emotion recognition and facial synkinesis *(Sunnybrook Facial Grading System: Synkinesis Score).* Moreover, in patients, there were significant negative correlations between accuracy for auditory emotion recognition and synkinesis of gentle eye closure, open mouth smile, and lip pucker. In the patients, there was a significant positive correlation between time for auditory emotion recognition and facial synkinesis (Sunnybrook Facial Grading System: Synkinesis Score). Moreover, in patients, there was a significant positive correlation between time for auditory emotion recognition and synkinesis in snarl.

#### 3.3.8. Accuracy/Time of Measured Facial Emotion recognition and Affected Side of Facial Synkinesis

Further, we performed *t*-tests to compare the facial emotion recognition of patients with left- and right-sided facial synkinesis.

A one-tailed *t*-test for independent samples for facial emotion recognition showed a trend for significance in accuracy between the patients with left-sided (Mean = 70.88 ± 8.25) and the patients with right-sided (Mean = 64.07 ± 13.45) facial synkinesis, t(28) = 1.694; *p* = 0.051. 

A one-tailed *t*-test for independent samples for facial emotion recognition demonstrated no significant difference in time between the patients with left-sided (Mean = 4.15 ± 0.54) and the patients with right-sided (Mean = 4.22 ± 1.02) facial synkinesis, t(28) = −0.398; *p* = 0.347. 

### 3.4. Self-Assessing Facial Emotion Recognition

Table 5 shows all the results of the self-assessed facial emotion recognition.

#### 3.4.1. Overall Competence

The *t*-test examining accuracy demonstrated no significant difference between the patients and the healthy controls, t(58) = −2.38; *p* = 0.406. Moreover, the *t*-test examining time demonstrated no significant difference between the patients and the healthy controls, t(58) = 0.523; *p* = 0.302. 

The *t*-tests within the groups demonstrated a significant higher self-assessment in accuracy, in comparison to time for the patients, t(29) = 1.894; *p* = 0.034. The same results go for the healthy controls, with a significant higher self-assessment in accuracy, in comparison to time, t(29) = 2.658; *p* = 0.006.

#### 3.4.2. Changes since Facial Palsy

The *t*-test examining accuracy demonstrated no significant difference between the patients and the healthy controls, t(58) = −1.134; *p* = 0.131. Moreover, the *t*-test examining time demonstrated no significant difference between the patients and the healthy controls, t(58) = −0.179; *p* = 0.429.

The *t*-test within the groups demonstrated no significant difference in accuracy, in comparison to time for the patients, t(29) = 1.179; *p* = 0.124. The same results go for the healthy controls with no significant difference in accuracy, in comparison to time, t(29) = 1.582; *p* = 0.062.

#### 3.4.3. Correlation between Accuracy and Time in Self-Assessed Facial Emotion Recognition

There was a significant correlation between self-assessed accuracy and time for facial emotion recognition in the patients, r = 0.829 (*p* < 0.001), and in the healthy controls, r = 0.653 (*p* < 0.001).

#### 3.4.4. Correlation between Accuracy and Time in Self-Assessed and Measured Facial Emotion Recognition

In the healthy controls, there was a significant correlation between self-assessed accuracy and measured accuracy for facial emotion recognition.

In the healthy controls, there was a significant correlation between self-assessed time and measured accuracy for facial emotion recognition (Table 6).

## 4. Discussion

The aim of this study is to observe the effects of post-paralytic facial synkinesis on facial emotion recognition. For this purpose, we examined patients with facial synkinesis and healthy controls in facial and auditory emotion recognition. The results of the standardised measurements as well as the self-reports present similar outcomes on facial and auditory emotion recognition between the groups of patients and healthy controls. Only in single cases, there were impairments, i.e., only limited performance, in facial and auditory emotion recognition in patients with pronounced facial asymmetry and facial synkinesis. Consequently, facial emotion recognition is a present competence of patients with facial synkinesis. The results will presently be discussed in more detail.

### 4.1. Comparison with Other Studies of Emotion Recognition and Facial Palsy

In a pilot study, Konnerth et al. [27], examined *n* = 13 patients with chronic peripheral facial palsy (sex: female 53.80%, male 46.20%; age: mean = 53.00 ± 17.64 years; facial palsy duration: mean = 7.82 ± 15.00 years). Facial emotion recognition resulted in average accuracy of 76.71 ± 12.60% and average time of 2.89 ± 0.95 s. By contrast, the patients in our study demonstrated lower accuracy and longer reaction times.

Storbeck et al. [26] studied *n* = 31 patients with acute peripheral facial palsy (sex: female 41.94%, male 58.06%; age: mean = 40.00 ± 2.3 years; facial palsy duration: mean = 7 ± 0.7 days and average moderate to moderately serve facial palsy). The facial emotion recognition resulted in an accuracy of 68.15 ± 2.25% and time of 10.77 ± 0.84 s (average data for test point t1). All the patients of our study demonstrated nearly similar performances in accuracy, but higher performance in time. 

For both previous studies, an ideal comparison with our study in facial emotion recognition cannot be conducted, because of the difference in the examination procedure and the sample compositions. Even so, since there are only a small number of studies examine emotion recognition in patients with facial palsy, the results should be considered [20]. Both studies used different emotion recognition tasks, whereby Konnerth et al. [27] is quite closer to the tools, used by us. The sex and age distribution differs as well as the time post onset facial palsy and its grading. Besides this incomparability between the studies, these may be factors that led to identification of deficits in facial emotion recognition time in contrast to healthy controls. In our study, we cannot confirm deficits in accuracy and time, but measured unimpaired facial and auditory emotion recognition. In future studies, facial emotion recognition should be tested with the same tool (1) at patients with acute and chronic facial palsy. This will make the results for different time stages post onset facial palsy comparable. It could be that facial emotion recognition is impaired in the early stage of acute facial palsy and recovers or could compensate in the chronic phase. In such an analysis, the typical course (paralytic, paretic, and possible synkinesis) [16,18] of facial palsy should be considered. Further on, (2) patients with acute and chronic flaccid paralysis with pronounced asymmetry should also be taken into account. Thus, the impact of facial asymmetry on facial emotion recognition can be investigated to a greater extent.

### 4.2. Emotion Recognition Depending on Facial Palsy

In line with Storbeck et al. (2019), our results reached the same conclusion regarding the correlation of facial emotion recognition and facial palsy. Storbeck et al. described no correlation between accuracy as well as time in facial emotion recognition and an overall grading of facial palsy [26], and neither did we. In total, this suggests the subordinate impact of facial feedback on facial emotion recognition. In particular, unilateral facial synkinesis, which may result in altered facial feedback, did not interrupt facial emotion recognition at first sight. But a detailed consideration of facial palsy provides new insights.

In contrast to Storbeck et al. (2019), who used the global *Facial Nerve Grading System 2.0* [46], we chose the comparatively finer-grained *Sunnybrook Facial Grading System* [33,34] for facial palsy grading. The separate assessment of resting symmetry, voluntary movements, and synkinesis in the *Sunnybrook Facial Grading System* [33,34] enables one to determine the significant correlation factors of asymmetry and synkinesis on facial and auditory emotion recognition in patients. The more asymmetric the face at rest, the less accurate was the measured facial emotion recognition. The more facial synkinesis, the less accurate and slower was the measured facial and auditory emotion recognition. For this, facial asymmetry may compromise individual patients in single cases in their accuracy of facial emotion recognition, especially for asymmetry in the mouth. Synkinesis may compromise patients’ facial and auditory accuracy and response time, especially for synkinesis in voluntary gentle eye closure, open mouth smile, snarl, and lip pucker. 

Synkinesis can lead to altered facial feedback, such as the so-called *autoparalytic syndrome*, in which different facial muscles interfere and inhibit each other [19]. The correlation between synkinesis and facial emotion recognition can be obviously explained by the facial feedback hypothesis, in which facial feedback is essential for successful facial emotion recognition [25]. While the correlation between synkinesis and auditory emotion recognition may be surprising at first, the effect of altered facial feedback in auditory emotion recognition is in line with previous research. Coles et al. (2019) cleared in their meta-analysis the significant effect of facial feedback and its dependence on the type of stimulus. While visual stimuli demonstrated a small effect, auditory stimuli was pointed out with high effects [47]. Thus, auditory emotion recognition can be affected by altered facial feedback [48]. Such as in prior research [48,49], our results may support evidence for facial feedback in overall emotion processing, and not only in facial emotion recognition, but also in auditory emotion recognition. In our sample, required facial feedback was still partially presented, e.g., because of unilateral (not bilateral) facial palsy. But in individual cases that particularly serve impairments in asymmetry and synkinesis, the facial feedback seemed to no longer be sufficient, and facial as well as auditory emotion recognition becomes affected.

Related to the side of the face, Korb et al. [28] reported an advantage in facial emotion recognition of patients with facial palsy on the right side in comparison to patients with facial palsy on the left side. We stated no significant difference in facial emotion recognition between patients with left- and right-sided facial synkinesis. However, in our data, there was a strong trend for a reversed effect. This aspect will thus need further attention in future research.

### 4.3. Comparison with Other Studies of Facial and Auditory Emotion Recognition

To test both facial and auditory emotion recognition within one study is rare in the previous literature [50]. Existing evidence demonstrates higher performance in facial than auditory emotion recognition [51,52,53]. In our study, the patients as well as the healthy controls were significantly more accurate but significantly slower in facial emotion recognition in comparison to auditory emotion recognition. Therefore, our results confirm existing evidence in accuracy while providing new evidence for a contrast in facial and auditory emotion recognition. For our sample (both patients and healthy controls), the statistical analysis revealed systematic significant correlations between facial and auditory emotion recognition. The more accurate the facial emotion recognition, the more accurate the auditory emotion recognition. The faster the facial emotion recognition, the faster the auditory emotion recognition. Thus, the ability to recognise emotions, regardless of modality, is more or less powerful within a person.

### 4.4. Correlations for Facial Emotion Recognition on Sex, Age, and Education

Additional factors demonstrated significant correlations on measured facial emotion recognition.

Among all participants, a correlation between accuracy and sex was noticeable. Women detected facially expressed emotions more accurately than men. These findings are consistent with previous research [29,30].

For all participants, the accuracy correlated significantly negative and the time correlated significantly positive with age. That means with increasing age, accuracy in facial emotion recognition decreased while time rises. Moreover, these results are in agreement with previous findings [31].

For all participants, there was a correlation between accuracy and education. This means, participants with higher education were more accurate in facial emotion recognition. Again, these findings are in line with previous research [32]. 

### 4.5. Self-Assessed Facial Emotion Recognition

We recorded self-assessed facial emotion recognition standardised in all participants. The more accurately participants (patients and healthy controls) assessed themselves in facial emotion recognition, the faster they assessed themselves. But only in the healthy controls, the more accurately and faster healthy controls assessed themselves in facial emotion recognition, the more accurate was the measured facial emotion recognition. This systematic significant correlation did not exist in patients. Thus, the patients’ self-assessment was less adequate compared to the measured emotion recognition.

### 4.6. Quality of Diagnostic Instruments

With our presented assessment of facial emotion recognition (cf. [21,22,23], we tested facial emotion recognition of patients with facial synkinesis. To construct different control parameters, we examined healthy controls without facial palsy and auditory emotion recognition as well. 

While we have uncovered impaired facial emotion recognition (accuracy) in patients with central facial palsy before [21,22,23], we revealed results regarding the abilities of patients with facial synkinesis. Besides the measurements, we took into consideration the participants‘ perspective with established (cf. [21,22,23], and newly developed questionnaires. Through the suggested measurement and self-assessment, we differentiated the participants (patients and healthy controls) and the patient groups (central and peripheral facial palsy) and detected deficits and competences. We also replicated expected correlation factors such as sex, age, and education, and new correlation factors such as asymmetry, facial synkinesis, and self-assessment, which further validates the quality of used diagnostic instruments.

### 4.7. Limitations of the Study

Emotions are usually not unimodal (facial or auditory), but multimodal [54]. A separation seems to be artificial, but enables us to declare the impact of facial feedback on facial emotion recognition. Since in this study the patients and healthy controls demonstrated no significant differences in facial emotion recognition, altered facial feedback (1) does not appear to have decisive influence or (2) is compensated already, where compensation by other modalities or context is excluded in this study design.

Furthermore, we solely tested basic emotions. In everyday life, more complex [38] and combinate [55] emotions have to be recognised in communication. Taking this perspective into consideration, the selection of basic emotions seems to be too experimental and unsuitable for everyday life. Further, the task of recognising basic emotions was maybe too basal, and deficits would become apparent in more complex and dynamic structures [56]. Or, it is the opposite and the stimuli were too obvious, and deficits in emotion recognition would even appear in slighter emotion expressions [49]. However, the choice of basic emotions is recommended [39] and that allows comparison with previous studies of patients with peripheral facial palsy (see above). Prospectively, performances of patients with central facial palsy should also be compared to reveal differences and parallels. After this research issue, the examination of facial emotion recognition should be improved through the expansion of multimodal and contextual information. Further studies should use more dynamic items and a variety of complex and combined emotions (c.f. [57,58]). All participants were examined, standardized via the *Sunnybrook Facial Grading System* [33,34], to rate their face (rest position, voluntary movements, and facial synkinesis) by a speech and language therapist. In future research, grading could be improved by machine learning approaches using facial landmarks [59,60,61,62] to be observer-independent [63]. An automatic assessment is of scientific value, because facial palsy and facial synkinesis as well as distinction between patients and healthy controls will become more standardised, objectified and thus more valid, reliable, and comparable within our study, and with other studies [60,61,62,64].

### 4.8. Consequences for the Care of Patients with Facial Synkinesis in Speech and Language Therapy

Up until now, high quality evidence in speech and language therapy for patients with facial palsy is rare but needed [65]. For evidence-based management, first, the impact of facial palsy and its consequences, such as facial synkinesis, have to be identified and described.

Our study introduced present competences in facial emotion recognition and call attention to the now uncovered risk of limitations in single cases of patients with pronounced facial asymmetry, and facial synkinesis as well as partially inadequate self-assessment. 

According to our results, the face of patients with facial palsy should be evaluated as a part of the standard clinical routine, for example by a speech and language therapist. In individual cases with pronounced facial asymmetry and serve facial synkinesis, facial and auditory emotion recognition should be quantified via objective measurements as well as with self-assessments of facial and auditory emotion recognition. Exclusively recording of self-assessment is not sufficient, as it may be inadequate. Any limitations indicate that therapy for emotion recognition is needed and should be offered. Next to speech and language therapy, psychology should be consulted as well. If there are no limitations but intact ability, facial emotion recognition should be considered as a resource of patients with facial synkinesis. The ability to understand facial emotion expressions may support the already affected verbal and nonverbal communication of these patients and their conversation partners. The intact facial emotion recognition should be integrated and strengthened in speech and language therapy. 

## 5. Conclusions

Facial emotion recognition is a present competence of patients with post-paralytic facial synkinesis. This resource should be integrated and strengthened in facial therapy. In future studies, the performance in the facial emotion recognition of patients with pronounced facial asymmetry in facial paralysis as well as central and peripheral facial palsy should be compared with objective instrument-based and also with standardised self-assessment tools in order to reveal differences and parallels.

## Figures and Tables

**Figure 1 diagnostics-12-01138-f001:**
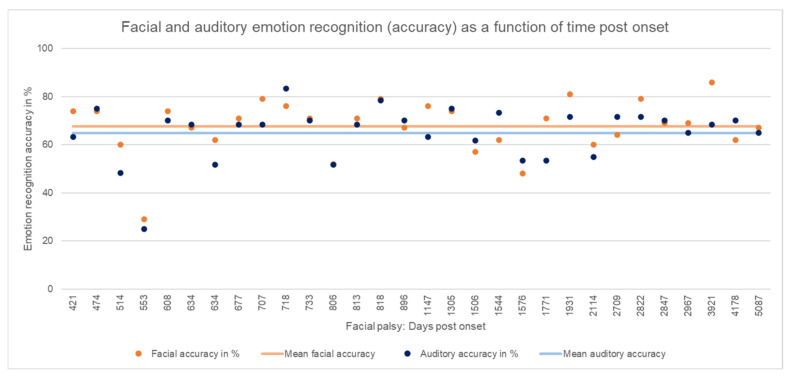
Facial and auditory emotion recognition (accuracy) as a function of time post onset. Individual results of the patients with facial synkinesis are shown as dots in bright orange (facial accuracy) and bright blue (auditory accuracy). Mean of facial accuracy in the patient group is shown as a line in pastel orange, and mean of auditory accuracy in the patient group is shown as a line in pastel blue.

**Figure 2 diagnostics-12-01138-f002:**
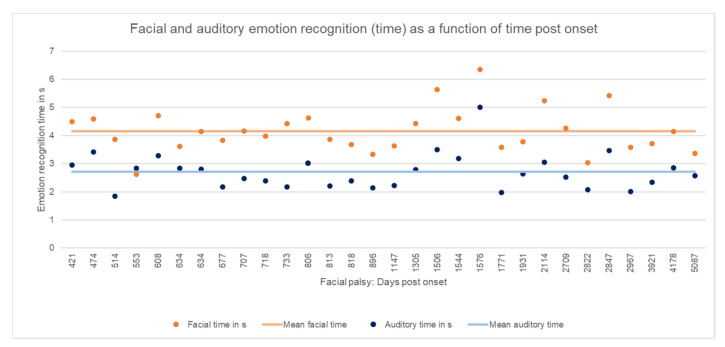
Facial and auditory emotion recognition (time) as a function of time post onset. Individual results of the patients with facial synkinesis are shown as dots in bright orange (facial time) and bright blue (auditory time). Mean of facial accuracy in the patient group is shown as line in pastel orange, and mean of auditory accuracy in the patient group is shown as a line in pastel blue.

**Figure 3 diagnostics-12-01138-f003:**
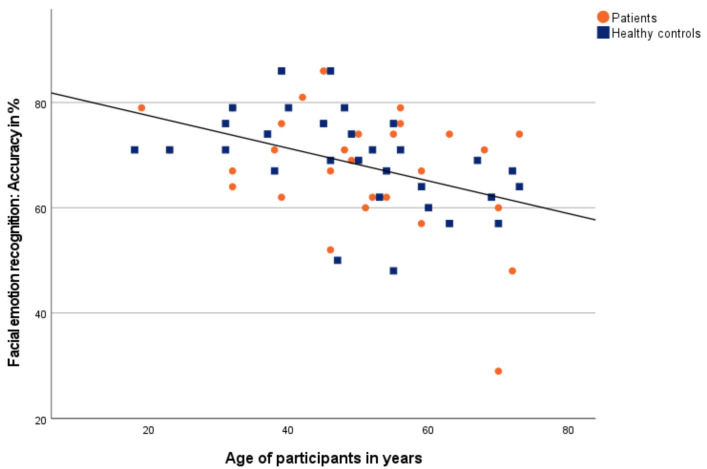
Correlation between accuracy of facial emotion recognition and age. Patients are shown as orange dots, and healthy controls as blue squares.

**Figure 4 diagnostics-12-01138-f004:**
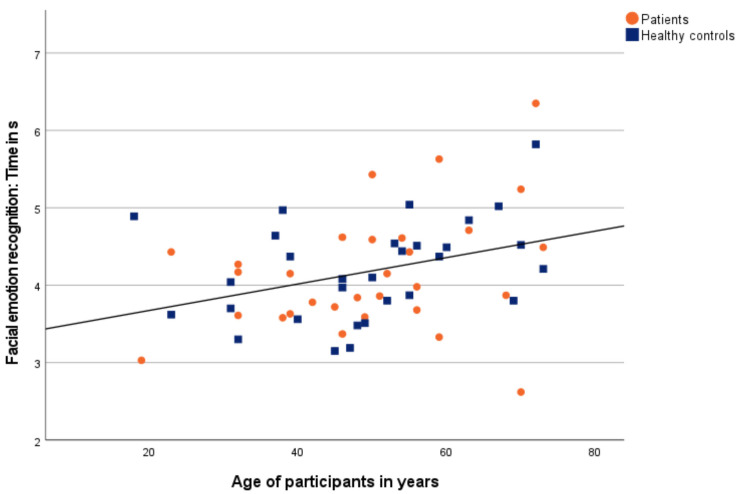
Correlation between time of facial emotion recognition and age. Patients are shown as orange dots, and healthy controls as blue squares.

**Figure 5 diagnostics-12-01138-f005:**
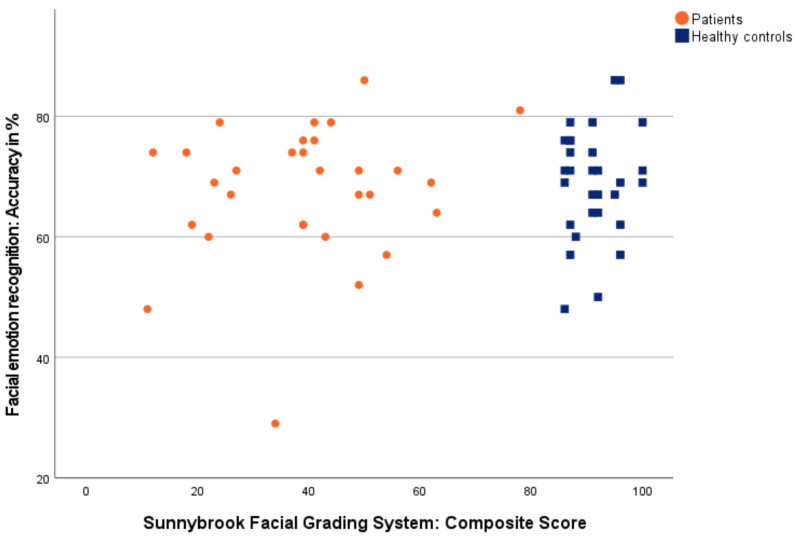
Correlation between accuracy of facial emotion recognition and grading of facial palsy. Patients are shown as orange dots, and healthy controls as blue squares.

**Figure 6 diagnostics-12-01138-f006:**
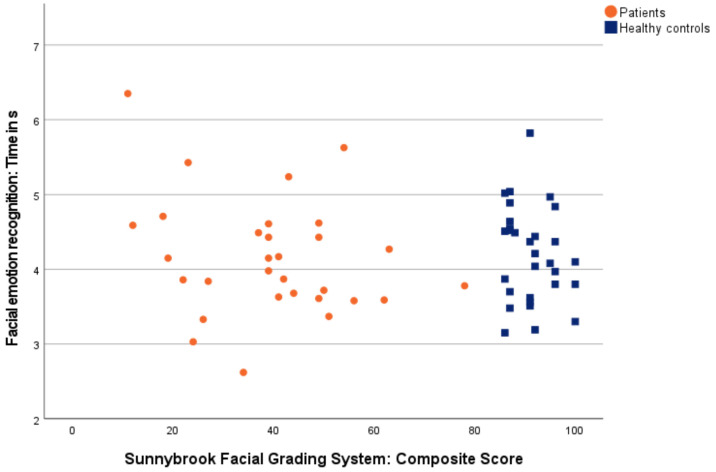
Correlation between time of facial emotion recognition and grading of facial palsy. Patients are shown as orange dots, and healthy controls as blue squares.

**Table 1 diagnostics-12-01138-t001:** Results of measured facial emotion recognition (FER)-accuracy/time, and auditory emotion recognition (AER)-accuracy/time.

Measured Emotion Recognition	Patients *n* = 30	Healthy Controls *n* = 30	
	Mean ± SD Min, Max	Mean ± SD Min, Max	
**FER Accuracy** via MFT Program in %	67.7 ± 11.3 Min 29 Max 86	69.1 ± 9.2 Min 48 Max 86	
**FER Time** via MFT Program in seconds	4.2 ± 0.8 Min 2.6 Max 6.4	4.2 ± 0.6 Min 3.2 Max 5.8	
**AER Accuracy** via MAV in %	64.9 ± 11.3 Min 25 Max 83.3	65.6 ± 10.3 Min 45 Max 80	
**AER Time** via MAV in seconds	2.7 ± 0.6 Min 1.9 Max 5	2.8 ± 0.7 Min 1.8 Max 5	
**Statistical analysis**	**Main effect of** ** *modality* **	**Main effect of** ** *group* **	**Interaction effect of** ** *modality × group* **
	**2 × 2 ANOVA** ** *p* **	**2 × 2 ANOVA** ** *p* **	**2 × 2 ANOVA** ** *p* **
**Accuracy**	F (1; 58) = 7.387 *p* = 0.009	F (1; 58) = 0.170 *p* = 0.682	F (1; 58) = 0.093 *p* = 0.762
**Time**	F (1; 58) = 441.501 *p* < 0.001	F (1; 58) = 0.170 *p* = 0.682	F (1; 58) = 0.219 *p* = 0.641

**Table 2 diagnostics-12-01138-t002:** Correlations between facial emotion recognition (FER)-accuracy/time, and auditory emotion recognition (AER)-accuracy/time.

Correlations	FER Accuracy		FER Time	
	Pearson r	*p*	Pearson r	*p*
**FER Time** All participants Patients Healthy controls	−0.150 −0.167 −0.131	0.126 0.189 0.245		
**AER Accuracy** All participants Patients Healthy controls	**0.650** **0.781** **0.473**	**<0.001** **<0.001** **0.004**		
**AER Time** All participants Patients Healthy controls			**0.712** **0.788** **0.638**	**<0.001** **<0.001** **<0.001**

Significant correlations are marked bold.

**Table 3 diagnostics-12-01138-t003:** Correlations between facial emotion recognition (FER)-accuracy/time, and sex, age, and education.

Correlations	FER Accuracy		FER Time	
	Pearson r	*p*	Pearson r	*p*
**Sex** All participants Patients Healthy controls	**−0.220** −0.221 −0.222	**0.046** 0.121 0.119	0.134 0.187 0.071	0.153 0.161 0.356
**Age** All participants Patients Healthy controls	**−0.427** **−0.398** **−0.468**	**<0.001** **0.015** **0.005**	**0.339** **0.340** **0.345**	**0.004** **0.033** **0.031**
	**Spearman ρ**	** *p* **	**Spearman ρ**	** *p* **
**Education** All participants Patients Healthy controls	**0.291** 0.213 **0.367**	**0.012** 0.129 **0.023**	−0.067 −0.029 −0.102	0.304 0.440 0.296

Significant correlations are marked bold.

**Table 4 diagnostics-12-01138-t004:** Correlations between facial emotion recognition (FER)-accuracy/time, and auditory emotion recognition (AER)-accuracy/time, and resting facial symmetry, and facial synkinesis.

Correlations	FER Accuracy	FER Time	AER Accuracy	AER Time
	Pearson r *p*	Pearson r *p*	Pearson r *p*	Pearson r *p*
**Composite Score** All participants	0.127 0.168	−0.062 0.320	0.072 0.293	−0.032 0.405
	**Spearman ρ** ** *p* **	**Spearman ρ** ** *p* **	**Spearman ρ** ** *p* **	**Spearman ρ** ** *p* **
**Resting Symmetry Score** Patients Healthy controls	**−0.441** **0.007** 0.163 0.195	0.051 0.304 0.151 0.213	−0.103 0.295 0.096 0.307	0.187 0.161 0.120 0.264
**Resting Symmetry Eye** Patients Healthy controls	−0.240 0.100 0.265 0.079	0.093 0.313 -0.041 0.414	−0.124 0.257 0.166 0.190	0.108 0.285 0.274 0.072
**Resting Symmetry Cheek** Patients Healthy controls	−0.041 0.414 0.076 0.344	0.181 0.169 0.084 0.330	−0.218 0.123 0.124 0.256	0.062 0.372 −0.148 0.218
**Resting Symmetry Mouth** Patients Healthy controls	**−0.353** **0.028** −0.106 0.288	0.087 0.324 0.164 0.194	−0.058 0.380 −0.184 0.166	0.029 0.440 0.091 0.315
**Voluntary Movement Score** Patients Healthy controls	−0.013 0.473 0.223 0.118	−0.139 0.232 −0.166 0.190	−0.079 0.338 0.205 0.138	−0.165 0.192 −0.048 0.400
**Synkinesis Score** Patients	**−0.348** **0.030**	0.267 0.077	**−0.474** **0.004**	**0.334** **0.035**
**Synkinesis in Brow lift** Patients	−0.234 0.107	0.038 0.421	−0.295 0.057	0.093 0.313
**Synkinesis in Gentle eye closure** Patients	0.071 0.355	− 0.210 0.133	**−0.383** **0.018**	0.020 0.457
**Synkinesis in Open mouth smile** Patients	−0.298 0.055	0.263 0.080	**−0.326** **0.040**	0.267 0.077
**Synkinesis in Snarl** Patients	**−0.334** **0.036**	**0.441** **0.007**	−0.244 0.097	**0.394** **0.016**
**Synkinesis in Lip pucker** Patients	−0.218 0.124	0.221 0.120	**−0.484** **0.003**	0.227 0.114

Significant correlations are marked bold.

**Table 5 diagnostics-12-01138-t005:** Results of self-assessed facial emotion recognition (accuracy and time).

Facial Emotion Recognition: Self-Assessment Questionnaire	Patients *n* = 30	Healthy Controls *n* = 30
	Mean ± SD Min, Max	Mean ± SD Min, Max
Overall competence: Accuracy	43.4 ± 7.5 Min 22.6 Max 54.4	43.8 ± 5.9 Min 26.1 Max 51.9
Overall competence: Time	41.7 ± 8.7 Min 16.7 Max 53.4	40.5 ± 8.9 Min 19.6 Max 53.5
Changes: Accuracy	0.1 ± 1.6 Min −4 Max 6	0.5 ± 1.1 Min −2 Max 3
Changes: Time	−0.2 ± 2.1 Min −6 Max 6	−0.1 ± 2.3 Min −6 Max 5

**Table 6 diagnostics-12-01138-t006:** Correlations between measured and self-assessed facial emotion recognition (FER)-accuracy/time.

Correlations	Measured FER Accuracy		Measured FER Time	
	Pearson r	*p*	Pearson r	*p*
**Self-assessed FER Accuracy** Patients Healthy controls	0.182 **0.438**	0.168 **0.008**	0.201 −0.090	0.144 0.318
**Self-assessed FER Time** Patients Healthy controls	0.268 **0.381**	0.076 **0.019**	0.196 −0.030	0.150 0.438

Significant correlations are marked bold.

## Data Availability

The data presented in this study are available on request from the corresponding author. The data are not publicly available due to the data being collected within a large research project that has not yet been completed.

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
