# Peer review of "Facial Emotion Recognition in Patients with Post-Paralytic Facial Synkinesis—A Present Competence"

_diagnostics, 2022, doi:10.3390/diagnostics12051138_

Round 1
Reviewer 1 Report
Overall the quality is good. I only have a minor suggestion. Authors a may need to discuss the state of the arts of facial analysis using AI in the related works. Such as:Robust Face Alignment by Multi-order High-precision Hourglass Networks
Author Response
Response to Reviewer 1 Comments
Point 1: Overall the quality is good. I only have a minor suggestion. Authors a may need to discuss the state of the arts of facial analysis using AI in the related works. Such as: Robust Face Alignment by Multi-order High-precision Hourglass Networks.
Response 1:
Thank you very much for your review and suggestion about AI.
We added a new paragraph and new literature as you recommended in the Discussion, section Limitaitons of the study, lines 621-629:
“All participants were examined standardised via the Sunnybrook Facial Grading System [33] [34] to rate their face (rest position, voluntary movements, and facial synkinesis) by a speech and language therapist. In future research, grading could be improved by machine learning approaches using facial landmarks [60] [61] [62] [63] to be observer-independent [64]. An automatic assessment is of scientific value, because facial palsy and facial synkinesis as well as distinction between patients and healthy controls will become more standardised, objectified and thus more valid, reliable, and comparable within our study, and with other studies [61] [62] [63] [65].”

Reviewer 2 Report
Aim of this study was to investigate the effects of post-paralytic facial synkinesis on facial and auditory emotion recognition. Results showed no differences between the performance, as well as the self-reports, of patients and healthy controls, thus, suggesting unimpaired ability of patients with facial palsy for emotion recognition. The study is interesting, however, there are some issues that need to be addressed.
Lines 119-125
The authors write that the patients and the healthy controls were matched in pairs based on their sex, age, and education level. Thus, based on the recruitment rules, it is obvious that there are no significant group differences between patients and healthy controls in sex, age and education level and I do not understand the meaning of the statistical analysis. Accordingly, I suggest removing statistic results also from Table 1. The same for Diagnosis of facial palsy in Table 2.
Table 2, Affected side. Please explain in the text or in the table notes what the statistical analysis refers to.
In Table 2. What these scores refer to? I do not see any explanation in the text.
Also, I do not think useful the statistical comparison between patients and controls for the grades I-VI (all healthy controls being healthy are grade I). On the contrary, it could be useful to show the differences in the grade's distribution for patients. In fact, about half patients are classified as Grade IV.
Lines 226-229.
As regards the correlation between accuracy/time and sex, age, education, severity and the 3 sub scores, did the authors do any correction (for example Bonferroni) for multiple correlations? I suggest the authors to perform it or to explain the reason for not doing it.
Lines 232-234
Why did the authors apply separate ttests instead of a more appropriate 2X2 ANOVA with group (patients vs. healthy controls) as between-subject factor and accuracy and time as within-subject repeated-measures factors?
3.1 Accuracy and 3.2 Time
For both accuracy and time, statistical analysis yielded no significant GroupxModality interactions, thus post hoc comparisons are not allowed.
Thus, the following paragraphs must be removed:” The post-hoc t-test confirmed that patients were significantly more accurate in facial 250 emotion recognition in comparison to auditory emotion recognition, t(29)=2.014; p=0.027. 251 Also, the healthy controls were significantly more accurate in facial emotion recognition 252 than in auditory emotion recognition, t(29)=1.888; p=0.035.”
“The post-hoc t-test within a group confirmed that patients are significantly slower in 277 facial emotion recognition compared to auditory emotion recognition, t(29)=16.399; 278 p<0.001. Also, the healthy controls were significantly slower in facial emotion recognition 279 than in auditory emotion recognition, t(29)=13.592; p<0.001.”
Figure 1 and 2
I think the captions should be modified.
For instance, it should be added that “lines” indicate the mean accuracy/time scores for facial (orange) and auditory (blue) emotion recognition.
Accordingly, inside the figures “facial mean of patients with facial palsy” and “auditory mean of patients with facial palsy” should be modified as, for example “mean facial accuracy/time” and “mean auditory accuracy/time”
As concerns analysis of the possible correlations between the various scores, I understand that the aim is “exploratory” but anyway I think that some of the presented correlations are seemingly nonsense, unless otherwise justified. For instance, what is the rationale for studying the correlation between accuracy for facial emotion recognition and time for auditory emotion recognition or between time for facial emotion recognition and accuracy for auditory emotion recognition?
I think the authors should briefly address these points also in the Introduction.
Table 4: I think the heading should be simplified. I suggest removing “measured” and using abbreviations. For instance, FER-accuracy/time and AER-accuracy/time.
Lines 320-321 what does it mean that there was a significant negative correlation between accuracy and sex across all participants (in favour of females)? Did you perform separate correlations between accuracy and females and males?
Throughout figures 3-6 I suggest making the captions easier. For instance, I suggest to use the terms Patients and Healthy controls instead of Patients with facial palsy and Healthy controls without facial palsy. The study is on patients with facial palsy (other patients are not included) compared with healthy controls, which being “healthy” cannot be affected by any disease.
Table 7. Please simplify the table. For example, you could organize the table as follows:
|
Facial emotion recognition: Self-Assessment Questionnaire |
Patients …... |
Healthy Controls |
|
Overall competence: accuracy |
|
|
|
Overall competence: time |
|
|
|
Changes: accuracy |
|
|
|
Changes: time |
|
|
Line 447. What does it mean there is a risk for limitations. What the authors found is a more marked impairment in patients with more pronounced asymmetry and synkynesis. Please clarify.
In the Discussion the authors compare their accuracy and time results for facial emotion recogniton with norm data. How can they assume that performance in accuracy is like norm data while performance in time is “lower” than norm data without a statistical analysis? The same for auditory modality.
Anyway, correct the following sentence at line 459 because it is nonsense:
“The better performances in facial time and auditory accuracy may have caused the younger sample from norm data”.
Author Response
Response to Reviewer 2 Comments
Aim of this study was to investigate the effects of post-paralytic facial synkinesis on facial and auditory emotion recognition. Results showed no differences between the performance, as well as the self-reports, of patients and healthy controls, thus, suggesting unimpaired ability of patients with facial palsy for emotion recognition. The study is interesting, however, there are some issues that need to be addressed.
Response: Thank you very much for your critical review. We welcome all your suggestions and addressed every point.
Point 1: Lines 119-125
The authors write that the patients and the healthy controls were matched in pairs based on their sex, age, and education level. Thus, based on the recruitment rules, it is obvious that there are no significant group differences between patients and healthy controls in sex, age and education level
and I do not understand the meaning of the statistical analysis. Accordingly, I suggest removing statistic results also from Table 1. The same for Diagnosis of facial palsy in Table 2.
Response 1:
The statistical analysis was conducted to demonstrate that the matching procedure had the intended effect: establish comparability of the two groups. We have followed your suggestion to remove the 2 tables from the body text and now present them in the Appendix A, Table A1 and Table A2.
Point 2: Table 2, Affected side
Please explain in the text or in the table notes what the statistical analysis refers to.
Response 2:
We removed Table 2 from the body text and now present it in the Appendix A, Table A2 (see also Response 1).
The statistical analysis for affected side showed, that there is no significant difference between the number of patients with left sided and right sided facial palsy. We added this information and explanation in the text: “About half of the patients were affected on the left side, while the other half
of the patients were affected on the right side of their face”, lines 158-159.
Point 3: In Table 2
What these scores refer to? I do not see any explanation in the text.
Response 3:
The Composite Score, and the sub scores Resting Symmetry Scores, Voluntary Movement Score, and Synkinesis Score belonging the the Sunnybrook Facial Grading System. We described it clearer now: “This tool is used to determine a composite total score (Composite Score; 0: total facial paralysis to 100: normal face) from sub scores for symmetry at rest (Resting Symmetry Score), voluntary
movements (Voluntary Movement Score), and synkinesis (Synkinesis Score) [33] [34].”, lines 141-145.
Point 4:
Also, I do not think useful the statistical comparison between patients and controls for the grades I-VI (all healthy controls being healthy are grade I). On the contrary, it could be useful to show the differences in the grade's distribution for patients. In fact, about half patients are classified as Grade IV.
Response 4:
We removed Table 2 from the body text and now present it in the Appendix A, Table A2 (see also Response 1).
Now, we showed the differences in the grade’s distribution in the text, Materials and Methods, Diagnosis and grading of facial palsy and facial synkinesis, lines 152--154: “Facial palsy was excluded in all healthy controls (Composite Score: Mean=91.3±4.5; Grade Median=1). In all patients, facial synkinesis was confirmed (Composite Score: Mean=39.4±15.8). About half of the patients had
medium facial paresis (Grade Median=4).”
Point 5: Lines 226-229
As regards the correlation between accuracy/time and sex, age, education, severity and the 3 sub scores, did the authors do any correction (for example Bonferroni) for multiple correlations? I suggest the authors to perform it or to explain the reason for not doing it.
Response 5:
With correlation calculation between accuracy/time and sex, age, and education, we proofed our own quality of diagnostics: We were able to replicate existing evidence that physiological emotion recognition depends on impact factors sex, age, and education, thus indicating that the overall pattern of results is as expected. In fact, we did not expect to learn new insights here, but confirm that our results are in line with previous studies. In this way, our data quality is able to prove existing hypothesis. In Materials and Methods, section Participants we explained the three well-known influencing factors on emotion recognition sex, age, and education, lines 121-123.
Furthermore, the perspective taken in this paper is that of a clinician considering potentially relevant factors for the intervention. Taking into account the three sub scores, statistical analysis is definitely exploratory. If we would use a Bonferonni-correctur in these correlation calculations, α-error would be minimized, of course, but β-error will increase. If we permit a minimal α-error, influencing factors, which are systematically related to the performance of emotion recognition, may results in no significant calcucation and remain overlooked. If we permit a higher β-error, significant effects may
no be detected, altough they are existing. For the sake of improving care of patients with facial synkinesis, it is better to potentially falsely detect a factor than to overlook an existing factor. For this, we decided to do no Bonferroni-correctur. In this way, we recommended in the Discussion, section Consequences for the care of patients with facial synkinesis in speech and language therapy, lines 641-649: “In individual cases with pronounced facial asymmetry and serve facial synkinesis, facial and auditory emotion recognition should be quantified via objective measurements as well as with self-assessments of facial and auditory emotion recognition. (…) If there are no limitations, but intact ability, facial emotion recognition should be considered as a resource of patients with facial synkinesis.”
Point 6: Lines 232-234
Why did the authors apply separate ttests instead of a more appropriate 2X2 ANOVA with group (patients vs. healthy controls) as between-subject factor and accuracy and time as within-subject repeated-measures factors?
Response 6:
We did 2 x 2 ANOVAs for measured facial and auditory emotion recognition, presented seperatly by accuracy and time, with group (patients vs. healthy controls) as between-subject factor and modality (facial vs. auditory) as within-subject repeated-measures factor.
For the facial self-assessment of patients and healthy controls, a 2 x 2 ANOVA is not possible:
Accuracy and response time are dependent variables, not independent variables, and thus do not constitute a factor for ANOVA. Because accuracy and respone time are not on the same sclae, a ANOVA calcucation is not possible and appropriate.
Point 7: 3.1 Accuracy and 3.2 Time
For both accuracy and time, statistical analysis yielded no significant GroupxModality interactions, thus post hoc comparisons are not allowed.
Thus, the following paragraphs must be removed:” The post-hoc t-test confirmed that patients were significantly more accurate in facial 250 emotion recognition in comparison to auditory emotion recognition, t(29)=2.014; p=0.027. 251 Also, the healthy controls were significantly more accurate in
facial emotion recognition 252 than in auditory emotion recognition, t(29)=1.888; p=0.035.”
“The post-hoc t-test within a group confirmed that patients are significantly slower in 277 facial emotion recognition compared to auditory emotion recognition, t(29)=16.399; 278 p<0.001. Also, the healthy controls were significantly slower in facial emotion recognition 279 than in auditory emotion
recognition, t(29)=13.592; p<0.001.”
Response 7:
3.1 Accuracy: We deleted the requested paragraph, lines 252-255.
3.2 Time: We deleted the requested paragraph, lines 278-281.
Point 8: Figure 1 and 2
I think the captions should be modified.
For instance, it should be added that “lines” indicate the mean accuracy/time scores for facial (orange) and auditory (blue) emotion recognition.
Accordingly, inside the figures “facial mean of patients with facial palsy” and “auditory mean of patients with facial palsy” should be modified as, for example “mean facial accuracy/time” and “mean auditory accuracy/time”
Response 8:
We added the indication of the lines in the figure notes:
“Mean of facial accuracy in the patient group is shown as line in pastel orange, and mean of auditory accuracy in the patient group is shown as a line pastel blue.” Lines 272-274 for Figure 1, and “Mean of facial accuracy in the patient group is shown as line in pastel orange, and mean of auditory accuracy in the patient group is shown as line in pastel blue.” Lines 293-294 for Figure 2.
Also as recommended, we replaced “facial mean of patients with facial palsy” and “auditory mean of patients with faical palsy” with “mean facial accuracy” and “mean auditory accurcy” in Figure 1.
We replaced “facial mean of patients with facial palsy” and “auditory mean of patients with facial palsy” with “mean facial time” and “mean auditory time” in Figure 2.
Point 9:
As concerns analysis of the possible correlations between the various scores, I understand that the aim is “exploratory” but anyway I think that some of the presented correlations are seemingly nonsense, unless otherwise justified. For instance, what is the rationale for studying the correlation between accuracy for facial emotion recognition and time for auditory emotion recognition or between time for facial emotion recognition and accuracy for auditory emotion recognition?
I think the authors should briefly address these points also in the Introduction.
Response 9:
We deleted the correlation between accuracy for facial emotion recognition and time for auditory emotion recogntion, and between time for facial emotion recognition and accuracy for auditory emotion recognition, Table 4.
Point 10: Table 4
I think the heading should be simplified. I suggest removing “measured” and using abbreviations.
For instance, FER-accuracy/time and AER-accuracy/time.
Response 10:
We used the word “measured” to differentiate between the reults of the emotion recognition tasks (measured), and the self-assessments of the participants. Now, we removed “measured” and used the suggested abbreviations in Table 2, and in its table description, lines 319-320.
To be consistent in the manuscript, now we use the suggested abbreviations in the other tables and removed “measured” in the table headings, too.
Point 11: Lines 320-321
What does it mean that there was a significant negative correlation between accuracy and sex across all participants (in favour of females)? Did you perform separate correlations between accuracy and females and males?
Response 11:
We performed one correlation for all participants with all females and males. Sex is a binary variable and thus the analysis is done in analogy to the other correlations, but actually tests for sex differences. We described clearer now: “There was a significant negative correlation between accuracy and sex
across all participants with higher accuracy for females.”, lines 324-325.
Point 12: Throughout figures 3-6
I suggest making the captions easier. For instance, I suggest to use the terms Patients and Healthy controls instead of Patients with facial palsy and Healthy controls without facial palsy. The study is on patients with facial palsy (other patients are not included) compared with healthy controls, which being “healthy” cannot be affected by any disease.
Response 12:
We made the captions easier as recommended. In Figures 3-6, the terms “Patients” and “Healthy controls” are used now instead of “Patients with facial palsy” and “Healthy controls withouf facial palsy”. To be consistent in the manuscript, now we use “Patients” and “Healthy controls” in all the tables/figures, too.
Point 13: Table 7
Please simplify the table. For example, you could organize the table as follows:
Facial emotion recognition: Self-Assessment Questionnaire
Patients
…...
Healthy Controls
Overall competence: accuracy
Overall competence: time
Changes: accuracy
Changes: time
Response 13:
We simplified Table 7 as recommended.
Point 14: Line 447
What does it mean there is a risk for limitations. What the authors found is a more marked impairment in patients with more pronounced asymmetry and synkynesis. Please clarify.
Response 14:
We clarified, lines 454-459:
“The results of the standardised measurements as well as the self-reports present similar outcomes on facial and auditory emotion recognition between the groups of patients and healthy controls. Only in single cases, there were impairments, i.e. only limited performance, in facial and auditory emotion
recognition in patients with pronounced facial asymmetry and facial synkinesis.”
Point 15: In the Discussion the authors compare their accuracy and time results for facial emotion recogniton with norm data. How can they assume that performance in accuracy is like norm data while performance in time is “lower” than norm data without a statistical analysis? The same for auditory modality.
Response 15:
We deleted the section Comparision to norm data in the Discussion, lines 463-471.
Point 16: Anyway, correct the following sentence at line 459 because it is nonsense:
“The better performances in facial time and auditory accuracy may have caused the younger sample from norm data”.
Response 16:
We deleted the whole section Comparison to norm data (see Response 15) including this sentence, lines 470-471.

Round 2
Reviewer 1 Report
I have no further suggestions